# On the Possibilities of Critical Raw Materials Production from the EU's Primary Sources

**Ewa Lewicka ***, **Katarzyna Guzik and Krzysztof Galos**

Mineral and Energy Economy Research Institute, Polish Academy of Sciences, J. Wybickiego 7A,
31-261 Kraków, Poland; guzik@min-pan.krakow.pl (K.G.); krzysztof.galos@min-pan.krakow.pl (K.G.)
* Correspondence: lewicka@min-pan.krakow.pl

**Abstract:** Sufficient supplies of critical raw materials (CRMs) for rapidly developing technologies, e.g., Li-ion batteries, wind turbines, photovoltaics, digitization, etc., have become one of the main economic challenges for the EU. Due to growing import dependency and associated risk of supply disruptions of these raw materials from third countries, there is a need to encourage their domestic production. This is an important starting point for EU value chains crucial for the sustainable economic growth of the whole Union. This contribution has evaluated the possibilities of CRMs supply from the EU's primary sources. A three-step approach, including an assessment of CRMs' importance for the EU's economic growth, their significance in at least two of the three strategic industrial sectors (i.e., renewable energy, e-mobility, defense and aerospace), and their potential availability from EU mineral deposits, has been applied. Results of the analysis have shown that, of 29 critical mineral raw materials (according to the 2020 EC list), the potential to develop manufacturing from the Union mineral deposits exists for 11 CRMs, i.e., cobalt, graphite (natural), HREE, LREE, lithium, magnesium, niobium, PGMs, silicon metal, titanium, and tungsten, while some other CRMs, namely gallium, germanium, indium, and vanadium can be recovered as by-products. Measures to mitigate EU import dependency have been also proposed.

**Keywords:** critical raw materials; mineral deposits; the EU; supply risk; value chains; emerging technologies; strategic industrial sectors

## 1. Introduction

Nowadays, technological progress and quality of life rely on access to numerous raw materials, the majority of which are closely linked to clean technologies, e.g., the production of PV panels, wind turbines, electric vehicles, and energy-efficient lighting. In order to maintain a steady, adequate, and sustainable supply of minerals, the appropriate policies should be implemented at all levels of government administration [1]. This particularly concerns mineral raw materials used in high-tech applications, e.g., electronics, digital technologies, robotics, or defense industry, which are essential for further industrial development [2–5].

Rapid economic development, in combination with accelerating technological innovations, has resulted in extreme changes in demand for numerous metals and minerals. This, coupled with an overall shift in the raw material markets into Asia (especially to China), has led to growing concerns about the security of their supply in industrialized economies, including the EU, justifying the urgent need for implementation of strategies with regard to mineral raw materials [6,7]. Strategies undertaken by various regions tend to differ in their foci. Whereas Europe opts for a dialogue with resource-rich countries, Japan and the United States have a more hands-on approach in research and development initiatives. However, Australia and China focus on the development of domestic mining activities and on resource protection [7].

Achieving resource security is a key question for the economic development and objectives of the EU climate policy, including the EU Green Deal Communication [8],

adopted on 11 December 2019. The European Union aspires to reduce the import dependency of raw materials that are critical for its industries by, among other goals, improving access to and utilization of their existing primary resources and increasing recycling activities [9–13]. Simultaneously, a strong focus has been put on the key elements of sustainable development—environmental, economic, and societal development [14,15].

Different countries have used various terms to define and identify minerals of economic importance, usually called critical or strategic raw materials (e.g., [16–28]). Distinctive lists of critical (or strategic) raw materials have also been created in individual countries, among which the United States pioneered the modern conceptualization of strategic minerals [29–32]. The latest US report [29,31] included three complementary indicators: the risk of supply disruption, the production growth rate, and the market development rate, which were applied to assess 78 non-energy minerals essential for defense applications. One of the main parameters used to determine the geopolitical risk of supply disruptions was the Herfindahl-Hirschmann Index (HHI) [31]. Interesting solutions have also been proposed in the United Kingdom [33,34] and Japan [35]. In the UK, a primary focus was placed on the risk of supply disruptions of 41 elements or their groups, while the issues related to their economic importance were disregarded. The main criteria adopted for the risk assessment were [17]: the concentration of production, distribution of resources, recycling scale, degree of substitution, and political stability indicators. In Japan, the latest criticality assessment was carried out for 39 high-tech metals in 5 categories: supply disruption risks, price risks, demand risks, recycling restrictions, and other potential risks. It should be also noted that the majority of the CRMs selection methods quantify supply risks and vulnerabilities in one base year, disregarding temporal changes; although, other views on the criticality of raw materials, including expected future demand or historic market development, have been also suggested [36–38].

In the European Union, critical raw materials are identified on the basis of two parameters, economic importance (EI) and risk of supply disruption assessment (SR) expressed by a value of relevant indicators [23,24]. The EI refers to the importance of a material for the EU economy in terms of end-use applications and the value added of corresponding EU manufacturing sectors at the NACE classification. Its value is corrected by the substitution index related to technical and cost performance of the substitutes for individual applications. The calculation of SR is based on the concentration of primary supply from raw materials producing countries (HHI), considering their governance performance measured by the World Governance Indicators (WGI) and trade aspects [24]. Possible substitution and recycling are considered risk-reducing measures. A list of critical raw materials was established by the European Commission (EC) as a priority action of the 'EU Raw Materials Initiative', launched by the Commission in 2008 [39]. In 2011, for the first time, the European Commission identified 14 critical non-energy non-agricultural raw materials for which undistorted, diversified, and affordable supplies should be ensured for the EU manufacturing industry [40,41]. Since then, the list has been a subject of review and update every three years: in 2014 (20 critical raw materials, [42]), in 2017 (27 CRMs, [43]), and in 2020 (30 CRMs [44]). The raw materials that have been most recently identified as critical for the EU have met the following criteria: a minimum threshold value of 2.8 points in the case of economic importance (EI), and a minimum threshold value of 1.0 point in supply risk assessment (SR) [44,45].

Sufficient supplies of raw materials essential to strategic value chains have become one of the main economic challenges of the EU [4,7,46–48]. Growing demand stems mainly from the advancing digital revolution, emerging innovations, e-mobility and artificial intelligence technologies, and the global energy transition [49,50]. The EU dependency on external supplies of the raw materials needed for its industry and economy is best reflected by supply risk (SR) and import dependency (IR) parameters. The most striking cases are posed by REEs (98–99% coming from China), borates (98% from Turkey), niobium (85% from Brazil), platinum (71% from South Africa), and cobalt (68% from Congo, DR) [45,46].

By considering all of the factors mentioned above, this article aims to assess the potential of the EU for the production of critical raw materials from primary sources (mineral deposits) located within its borders. Contrary to EU criticality assessments, in this study, an economic importance analysis of the CRMs has been evaluated in relation to three selected industry sectors (i.e., renewable energy, e-mobility, defense and aerospace). These sectors are essential in the implementation of the EU's priorities in terms of transformation towards climate neutrality and digital leadership [8,51], as well as in raw materials security.

## 2. Materials and Methods

This paper evaluates mineral raw materials determined as critical for the EU, according to the fourth EC communication on critical raw materials [44] and the EC Study on the EU's list of Critical Raw Materials—Final Report [45], with regard to its existing resource base and the security of supply chains for its strategic industry sectors [24]. The basic sources of information were numerous studies [2,3,51–58] and results of a few EU-financed projects (e.g., [53,59–64]). The CRMs analyzed in this paper include 29 mineral raw materials (except for natural rubber, with it not being a mineral), i.e., antimony, baryte, bauxite, beryllium, bismuth, borates, coking coal, cobalt, fluorspar, gallium, germanium, hafnium, indium, lithium, magnesium, natural graphite, niobium, PGMs, phosphate rock, phosphorus, HREEs, LREEs, scandium, silicon metal, strontium, tantalum, titanium, tungsten, and vanadium.

Our assessment was carried out in three steps:

- The first step focused on CRMs' domestic and external sourcing, production and/or processing in the EU to manufacture a wide range of value-added products of key importance for the EU's economic growth, as well as on their key applications, especially with regard to new technologies (Table 1);
- The second step was related to the importance of CRMs for strategic technologies and sectors development, i.e., renewable energy, e-mobility, defense, and aerospace, that have been identified by the European Commission [51]; raw materials recognized as important for at least two strategic industry sectors were selected for further analysis in the third step (Table 2);
- In the third step the potential availability of the primary sources of the selected critical raw materials in the EU, including deposits with identified resources and/or reserves, as well as occurrences or showings, was examined and assessed on the basis of existing data collected from past EU-financed projects [53,54,59,60], as well as on the European Commission reports [40–43,46,47] (Table 3).

Table 1. An assessment of the critical raw materials importance for the EU economy [40–45,51].

| Mineral Raw Material | Classified as Critical Raw Material | | | | EI Index | SR Index | Mining Production in the EU | Processing in the EU | Production of Semi-Finished Products in EU | Subject of Trade Flows | Production in the EU | Main Applications |
|---|---|---|---|---|---|---|---|---|---|---|---|---|
| | 2011 | 2014 | 2017 | 2020 | 2020 | 2020 | | | | | | |
| Antimony | YES | YES | YES | YES | 4.8 | 2.0 | NO | YES [1] | YES | unwrought metal, antimony trioxide (ATO), antimony powders, scrap | antimony trioxide (ATO) | flame retardants, lead-acid batteries, lead alloys, plastics (catalysts and stabilizers), glass and ceramics |
| Baryte | NO | NO | YES | YES | 3.3 | 1.3 | YES | YES | YES | baryte aggregates, ground and micronized baryte, blanc baryte | ground baryte | weighting agent in drilling fluids, filler in rubbers, plastics, paints and paper, chemical industry |
| Bauxite | na | NO | NO | YES | 2.9 | 2.1 | YES | YES | YES | bauxite (dried or calcined) | bauxite (dried or calcined) | alumina (mostly for the production of aluminum metal), refractories, cement, abrasives, chemicals |
| Beryllium | YES | YES | YES | YES | 4.2 | 2.3 | NO | NO | YES | Be metal, Be alloys, and master alloys, Be oxides | - | electronics, automotive components, aerospace components, energy applications |
| Bismuth | na | na | YES | YES | 4.0 | 2.2 | NO | YES | YES | refined bismuth | refined bismuth, Bi chemicals, Bi alloys | chemicals, fusible alloys, metallurgical additives, and others |
| Borates | NO | YES | YES | YES | 3.5 | 3.2 | NO | YES | YES | natural borates, boric acid, boron metal | - | specialty glass and glass fiber, frits and ceramics, fertilizers, chemicals, construction materials, boron metal production |
| Cobalt | YES | YES | YES | YES | 5.9 | 2.5 | YES | YES | YES | ores and concentrates, oxides and hydroxides, chlorides, intermediate products, refined cobalt | refined cobalt | superalloys, hard-facing alloys, hard materials (carbides and diamond tools), pigments, catalysts, magnets, batteries |
| Coking coal | na | YES | YES | YES | 3.0 | 1.2 | YES | YES | YES | coking coal, coke | coking coal, coke | iron and steel production, tar and benzol production |
| Fluorspar | YES | YES | YES | YES | 3.3 | 1.2 | YES | YES | YES | fluorspar AG, fluorspar MG, cryolite, fluorine compounds | fluorspar AG (min. 97% $CaF_2$) | Fe making, Al making, UF6 in nuclear uranium fuel, HF in oil refining, CFCs for refrigeration and air conditioning |
| Gallium | YES | YES | YES | YES | 3.5 | 1.3 | NO [2] | YES | YES | unwrought gallium, galium compounds (e.g., GaAs) | refined high purity gallium | integrated circuits, electronics, LED lighting, CIGS (Cu-In-Se-Ga) photovoltaics panels |
| Germanium | YES | YES | YES | YES | 3.5 | 3.9 | NO | YES | YES | germanium metal and powders, $GeO_2$, $GeCl_4$ | germanium metal, $GeCl_4$ | IR optics, optical fiber, satellite solar |
| Graphite (natural) | YES | YES | YES | YES | 3.2 | 2.3 | YES | YES | YES | graphite powder, graphite flakes, other natural graphite | natural graphite concentrates | refractories, Li-ion and other types of batteries, friction products, lubricants, pencils |

**Table 1.** *Cont.*

| Mineral Raw Material | Classified as Critical Raw Material | | | | EI Index | SR Index | Mining Production in the EU | Processing in the EU | Production of Semi-Finished Products in EU | Subject of Trade Flows | Production in the EU | Main Applications |
|---|---|---|---|---|---|---|---|---|---|---|---|---|
| | 2011 | 2014 | 2017 | 2020 | 2020 | 2020 | | | | | | |
| Hafnium | na | NO | YES | YES | 3.9 | 1.1 | NO | YES [3] | YES | hafnium metal | hafnium oxide, hafnium metal | superalloys, nuclear energy production (nuclear control rods), semiconductors |
| Indium | YES | YES | YES | YES | 3.3 | 1.8 | NO | YES [4] | YES | In-bearing zinc concentrates, residues, and slags; unwrought indium | refined indium | flat monitors, CIGS (Cu-In-Se-Ga) photovoltaics panels, solders, batteries, semiconductors, and LEDs |
| Lithium | na | na | NO | YES | 3.1 | 1.6 | YES [5] | NO | NO | lithium carbonate and hydroxide, lithium spodumene concentrate | lepidolite concentrate, lithium metal | glass and ceramics, batteries, lubricants, aluminum production, pharmaceuticals |
| Magnesium | YES | YES | YES | YES | 6.6 | 3.9 | NO | NO | YES | magnesium metal, magnesium alloys | Magnesium alloys | automotive industry, steel desulfurization agent, packaging, construction |
| Niobium | YES | YES | YES | YES | 6.0 | 3.9 | NO | NO | YES | ferroniobium, niobium metal | Nb-based alloys, Nb chemicals (e.g., lithium niobate) | HSLA steel (for construction and vehicles), stainless and special steel, chemicals |
| Platinum Group Metals | YES | YES | YES | YES | 5.7 | 2.4 | YES | YES | YES | PGM metals, PGM alloys, PGM chemicals | concentrates, refined PGMs, alloys, PGM chemicals | catalysts (automotive, chemical and petroleum), electronics, glass, jewelry, dental, investment |
| Phosphate rock | na | YES | YES | YES | 5.6 | 1.1 | YES | YES | YES | phosphate rock, phosphoric acid, phosphate fertilizers | phosphate rock, phosphate acid, fertilizers | phosphate and multicomponent fertilizers, food additives, detergents, flame retardants |
| Phosphorus | na | na | YES | YES | 5.3 | 3.5 | NO | NO | YES | elemental phosphorus | phosphoric acid | chemicals, electronics, metals production |
| Rare Earth Elements | YES | YES | YES | YES | LREE 4.3 HREE 3.9 | LREE 6.0 HREE 5.6 | NO | YES | YES | REE oxides (REO), REE metals and alloys, REE compounds | REE chemicals and compounds, REE metals | catalysts, permanent magnets (for automotive applications), special alloys, glass, and ceramics, phosphors, batteries, electronics |
| Scandium | YES | NO | YES | YES | 4.4 | 3.1 | NO | YES | YES | scandium oxide (also scandium compounds and scandium metal) | Sc-Al alloys | Solid Oxide Fuel Cells (SOFC), Sc-Al alloys |

**Table 1.** *Cont.*

| Mineral Raw Material | Classified as Critical Raw Material | | | | EI Index | SR Index | Mining Production in the EU | Processing in the EU | Production of Semi-Finished Products in EU | Subject of Trade Flows | Production in the EU | Main Applications |
|---|---|---|---|---|---|---|---|---|---|---|---|---|
| | 2011 | 2014 | 2017 | 2020 | 2020 | 2020 | | | | | | |
| Silicon metal | na | YES | YES | YES | 4.2 | 1.2 | YES | YES | YES | silicon metal, intermediate products (Si-based chemicals, silicon wafers) | silicon metal | chemicals (silicons and silanes), aluminum alloys, semiconductors (photovoltaics, wind turbines, electronics), Li-ion batteries |
| Strontium | na | na | na | YES | 3.5 | 2.6 | YES | YES | YES | strontium ore and concentrates, strontium carbonate, strontium metal | celestite concentrate, strontium carbonate | glass, ceramics, pyrotechnics, magnets, master alloys, drilling fluid |
| Tantalum | YES | NO | YES | YES | 4.0 | 1.4 | NO [6] | YES | YES | tantalum pentoxide, tantalum ores and concentrates | tantalum chemical compounds, superalloys | capacitors, superalloys (aviation), carbides |
| Titanium | NO | NO | NO | YES | 4.7 | 1.3 | NO | YES [7] | YES | titanium ores and concentrates, titanium white, titanium metal | titanium white, titanium metal, alloys | alloys (space and aircraft, military, and medical applications), paints and polymers (plastics) |
| Tungsten | YES | YES | YES | YES | 8.1 | 1.6 | YES | YES | YES | tungsten ores and concentrates, tungsten carbides, powders, APT | ores and concentrates, APT | mill and cutting tools, other wear tools, catalysts, and pigments, lighting, electronics, aeronautics, and energy uses |
| Vanadium | NO | NO | YES | YES | 4.4 | 1.7 | NO | YES | YES | vanadium ores and concentrates, vanadium oxides, ferrovanadium | vanadium oxides, ferrovanadium | ferrovanadium, HSLA steel, Al-Ti-V alloys (aviation, nuclear energy), stainless and special steel, catalysts (chemical) |

Explanations: EI—economic importance; SR—supply risk; na—not assessed; [1] production of antimony trioxide on the basis of imported unwrought antimony; [2] currently, gallium is not produced in the EU; it was recovered until 2013 in Hungary and until 2016 in Germany in the course of metallurgical treatment of gallium-bearing bauxite; [3] hafnium is a by-product of zirconium ore processing; [4] indium is produced in the EU from imported In-rich zin concentrates and In-rich waste materials; Neves Corvo mine in Portugal produces In-rich Zn and Sn concentrates, but there is no information regarding whether indium is recovered there; [5] only ceramic grades; [6] tantalum is obtained in small quantities by Imerys in Echassieres (France), but the entire production is exported out of the EU; [7] titanium ores are sourced, e.g., from Norway.

**Table 2.** Critical raw materials required in the EU's strategic industrial sectors (according to [51]).

| CRMs | Renewable Energy | E-Mobility | Defense and Aerospace | CRMs in at Least Two Sectors |
|---|---|---|---|---|
| Antimony (Sb) | | | | |
| Baryte | | | + | |
| Bauxite | | | | |
| Beryllium (Be) | | | + | |
| Bismuth (Bi) | | | | |
| Borates (and boron) | + | + | + | + |
| Cobalt (Co) | | + | + | + |
| Coking coal | | | | |
| Fluorite | | | | |
| Gallium (Ga) | + | + | + | + |
| Germanium (Ge) | + | + | + | + |
| Graphite natural | | + | + | + |
| Hafnium (Hf) | | | + | |
| Indium (In) | + | + | + | + |
| Lithium (Li) | | + | + | + |
| Magnesium (Mg) | | + | + | + |
| Niobium (Nb) | + | + | + | + |
| Phosphate rock | | | | |
| Phosphorus | | | | |
| Platinum Group metals (PGMs) | | + | + | + |
| Heavy Rare Earth Elements (HREEs) | + | + | + | + |
| Light Rare Earth Elements (LREEs) | + | + | + | + |
| Scandium (Sc) | | | | |
| Silicon metal (Si) | + | + | | + |
| Tantalum (Ta) | | | + | |
| Titanium (Ti) | | + | + | + |
| Tungsten (W) | | + | + | + |
| Vanadium (V) | | + | + | + |

**Table 3.** The EU countries with deposits or mineral occurrences of critical raw materials required in at least two strategic industry sectors [7,10,25,27,33,39,44].

| Mineral Raw Material | Countries with Reported: | | | | Main Types of Deposits |
|---|---|---|---|---|---|
| | Mineral Deposits According to [60] | Mineral Resources According to [61] | Mineral Reserves According to [61] | Mineral Occurrences/Deposits According to [54] | |
| Borates | - | not known | not known | not analyzed | evaporites in volcanic activity areas, |
| Cobalt | Finland, Greece, Poland, Spain, Sweden | Finland, Germany, Sweden | Finland | Austria, Bulgaria, Cyprus, Czech Republic, Finland, France, Germany, Greece, Italy, Poland, Romania, Slovakia, Spain, Sweden | sediment-hosted, hydrothermal, and volcanogenic, magmatic sulphide deposits, laterites |
| Gallium [1] | Poland | data not available | data not available | Austria, Bulgaria, France, Hungary, Poland | rarely forms its own deposits, mostly occurring as a trace element in bauxite ores, subordinately in Zn ores |
| Germanium [2] | Austria, France, Slovenia, Poland | France, Czech Republic | not known | Austria, Bulgaria, Czech Republic, France, Germany, Italy, Poland, Portugal, Romania, Slovenia | does not form its own deposits; mostly occurring in as trace metal in Zn ores and coal ashes |
| Graphite natural | Austria, Bulgaria, Czech Republic, Germany, Sweden | Austria, Czech Republic, Germany, Spain, Slovakia, Sweden | Austria, Czech Republic, Spain | Austria, Czech Republic, Finland, France, Germany, Greece, Italy, Romania, Sweden | flake graphite, amorphous graphite, vein/lump graphite |
| Indium [3] | Germany, Portugal | Germany | data not available | Austria, Bulgaria, Czech Republic, Germany, Greece, Hungary, Ireland, Portugal | does not form its own deposits; occurs primarily as trace element in Zn ores |
| Lithium | Austria, France, Ireland, Slovenia | Austria, Czech Republic, Finland, France, Ireland, Germany, Portugal, Spain, Sweden | Austria, Czech Republic, Finland, Germany, Portugal | Czech Republic, Finland, France, Greece, Portugal, Spain, Sweden | pegmatite, brines, thermal water |
| Magnesium | Slovakia, Greece | Austria, Bulgaria, Greece, Ireland, Poland, Slovakia, Spain | Austria, Poland, Slovakia, Spain | Austria, Bulgaria, Finland, France, Greece, Ireland, Italy, Netherlands, Poland, Slovakia, Spain, United Kingdom * | high purity deposits of dolomite, magnesite and carnalite |
| Niobium [4] | Czech Republic | Finland, France, Germany, Portugal, Sweden | data not available | Austria, Bulgaria, Czech Republic, Finland, France, Germany, Italy, Portugal, Slovakia, Spain | carbonatite-hosted primary, carbonatite-sourced secondary, alkaline granite and syenite |
| Platinum Group Metals | Finland, Poland, Sweden | Finland, Germany, Sweden | Finland | Bulgaria, Finland, Greece, Germany, Poland, Spain, Sweden, United Kingdom * | PGM-bearing (Merensky Reef type and chromite reef type), nickel-copper sulfides |
| Rare Earth Elements (LREEs, HREEs) | Greece, Finland, Portugal, Sweden | Greece, Finland, Germany, Portugal, Sweden | Sweden | Belgium, Czech Republic, Finland, France, Greece, Germany, Poland, Portugal, Romania, Slovakia, Spain, Sweden, United Kingdom * | carbonatite-associated, laterite (ion adsorption deposits), alkaline igneous rock, placers |
| Silicon metal [5] | data not available | Czech Republic, Greece, Latvia, Poland, Slovakia, Slovenia, United Kingdom * | Croatia, Czech Republic, Denmark, Poland, Slovakia, Slovenia, United Kingdom * | Austria, Bulgaria, Finland, Greece, Germany, Poland, Portugal, Sweden, Italy | high purity silica sand, quartz veins, quartzites |

**Table 3.** *Cont.*

| Mineral Raw Material | Countries with Reported: | | | | Main Types of Deposits |
|---|---|---|---|---|---|
| | Mineral Deposits According to [60] | Mineral Resources According to [61] | Mineral Reserves According to [61] | Mineral Occurrences/Deposits According to [54] | |
| Titanium | Finland, France, Portugal, Romania | Finland, France, Portugal, Slovakia, Sweden | Slovakia | Finland, France, Greece, Italy, Portugal, Romania, Sweden | primary: igneous, metamorphic; weathered rocks and unconsolidated sediments (placers) |
| Tungsten | Austria, Czech Republic, Finland, France, Germany, Greece, Spain, Portugal, Sweden, United Kingdom * | Austria, Bulgaria, Czech Republic, Germany, Finland, Poland, Portugal, Slovakia, Spain, Sweden, United Kingdom * | Austria, Spain, United Kingdom * | Austria, Bulgaria, Czech Republic, Finland, France, Germany, Greece, Spain, Portugal, Sweden, United Kingdom * | vein/stockwork, greisen, hydrothermal, skarn |
| Vanadium | Finland, Poland, Sweden, United Kingdom * | Sweden | data not available | Estonia, Finland, Poland, Sweden | sedimentary phosphates, bauxites, fossil fuel |

Explanations: [1] Recovered exclusively as by-products during the processing of other metals (aluminum, zinc), occurring primarily from bauxite deposits, resources of gallium have not been evidenced in the EU, although there was some gallium production in the past in Hungary and Germany; [2] recovered mostly from zinc ores and coal ashes; [3] derived as by-product of zinc production; [4] data on resources and reserves reported together for niobium and tantalum; [5] data for high-quality silica sand/vein quartz; * United Kingdom had been a member state of the EU until 2019.

In the course of this analysis the EU's own potential primary sources of CRMs, essential for the strategic value chains and strategic industrial sectors, which could be utilized in the manufacturing industries in the EU, have been determined (Table 4).

**Table 4.** Selected critical mineral raw materials important for the EU's strategic industry sectors with the highest potential for developing the mining production from domestic resources [44,45,51,53].

| CRMs 2020 | EI 2020 | SR 2020 | IR (%) | Deposits * | Renewable Energy | E-Mobility | Defense and Aerospace |
|---|---|---|---|---|---|---|---|
| CRMs with the highest potential for mining production development from their own deposits | | | | | | | |
| Cobalt (Co) | 5.9 | 2.5 | 86 | + | | + | + |
| Graphite natural | 3.2 | 2.3 | 98 | + | | + | + |
| Lithium (Li) | 3.1 | 1.6 | 100 | + | | + | + |
| Niobium (Nb) | 6.0 | 3.9 | 100 | + | + | + | + |
| Platinum Group Metals (PGMs) | 5.7 | 2.4 | 98 | + [1] | | + | + |
| Heavy Rare Earth Elements (HREEs) | 3.9 | 5.6 | 100 | + | + | + | + |
| Light Rare Earth Elements (LREEs) | 4.3 | 6.0 | 100 | + | + | + | + |
| Titanium (Ti) | 4.7 | 1.3 | 100 | + | | + | + |
| Tungsten (W) | 8.1 | 1.6 | na | + | | + | + |
| CRMs with the highest potential for production development as by-products from other metal ores | | | | | | | |
| Gallium (Ga) | 3.5 | 1.3 | 31 | bauxite | + | + | + |
| Germanium (Ge) | 3.5 | 3.9 | 31 | Zn-Pb ore | + | + | + |
| Indium (In) | 3.3 | 1.8 | 0 | Zn, Zn-Cu-Sn ore | + | + | + |
| Vanadium (V) | 4.4 | 1.7 | na | Fe, Fe-Ti ore | | + | + |

Explanations: EI—economic importance; SR—supply risk; IR—import reliance; * own mineral deposits or deposits of other minerals or metals are sources for the recovery of the raw material (as a by-product); na—not available; [1] also from Cr and Ni-Cu ore deposits.

## 3. Results

### 3.1. Mineral Raw Materials of High Economic Importance for the EU, Their Sources, Criticality and Applications

According to the latest criticality assessments [44,45], an index of economic importance (EI) of CRMs generally ranged from 2.9 for bauxite to 8.1 for tungsten. The score of SR index changed in the range of 1–6 points, with the lowest values for phosphates and hafnium (1.1 points each) and the highest for rare earths elements (LREE—6.0, HREE—5.6).

A thorough analysis of these data showed that, among 29 analyzed CRMs (excluding natural rubber) only 12 were sourced—to a various degree—from deposits located within the EU. These are the following raw materials: baryte, bauxite, cobalt, coking coal, fluorite, graphite (natural), lithium, phosphates, platinum group metals (platinum and palladium), silicon metal, strontium, and tungsten (Table 1). The remaining CRMs were basically imported from non-EU countries, though some of them—e.g., bismuth, gallium, germanium, hafnium, indium, and tantalum—were to some extent recovered from various imported ore concentrates as by-products [65–67].

### 3.2. Assessment of Importance of CRMs for Strategic Technologies and Sectors Development

The risk of disruption to raw material supplies along the strategic supply chains in the EU has been carried out in detail in the 'EC foresight study' [52]. It focused especially on technologies related to renewable energy, e-mobility, and defense and aerospace applications [2,3,52]. The set of nine technologies with key CRMs required for these strategic industry sectors development include:

- Li-ion batteries (LiBs), which are emerging as an important technology across a wide range of civil and defense applications. As a result of the increasing spreading of electric vehicles, mobile electrical appliances, and stationary energy storage sys-

tems, the demand for LiBs is expected to skyrocket (>30% per year) for the next 10 years [4,51,68,69]. The basic CRMs required in this technology are cobalt, graphite (natural), lithium, niobium, silicon, and titanium.

- Fuel cells (FCs)—the deployment of FCs has grown during the last 10 years, but it is still uncertain when they will be widely commercialized. The main barriers to their widespread use are reliability (availability and lifetime), efficiency, and cost [51]. The CRMs essential for the production of fuel cells are cobalt, graphite (natural), palladium, platinum, titanium, strontium.

- Wind turbine generators are among the most cost-effective technologies in the clean energy generation in the EU [49]. The most relevant CRMs required include boron metal and borates, dysprosium, niobium, neodymium, and praseodymium. The main critical material containing components in wind turbines are the permanent magnets.

- Electric traction motors (permanent magnets) are also used in numerous applications for small electronic products, e-bikes, electric cars, and heavy transportation. In the future, NdFeB magnet technology is expected to dominate the market; by 2025, between 90% and 100% of hybrids and EVs could be driven by NdFeB-containing motors [49,51]. Critical raw materials utilized in traction motors are boron, dysprosium, neodymium, and praseodymium.

- Photovoltaics (PV), together with wind energy, are expected to lead the transformation of the global and the EU electricity sector [5,50]. The most common CRMs used in this technology include borates, gallium, germanium, indium, and silicon metal.

- Robotics is an emerging technology with enormous potential for many applications. Out of the 44 raw materials used in robotics, 19 materials are identified as critical for the EU [51]. The most important are boron, beryllium, dysprosium, gallium, indium, niobium, neodymium, praseodymium, and titanium.

- Drones are used for various civil and military applications. Of the 48 raw materials, 15 materials, namely borates, bismuth, beryllium, cobalt, gallium, graphite (natural), hafnium, indium, lithium, magnesium, niobium, PGMs, REEs, antimony, silicon metal, tantalum, titanium, tungsten, and vanadium, are identified as critical to the EU economy [2,51].

- 3D printing (3DP) technology utilization is expected to grow substantially, especially in aerospace, defense, and medical industries. However, key challenges include achieving sufficient quality and lowering the production cost [51]. The main CRMs required are cobalt, hafnium, magnesium, niobium, scandium, silicon metal, titanium, vanadium, and tungsten.

- Digital technologies are strategic technologies that have changed the contemporary style of life and communication, as well as industrial productivity [51]. The essential raw materials in these technologies include boron, cobalt, gallium, germanium, graphite, indium, lithium, magnesium, PGMs, REE, silicon metal, strontium, titanium, and tungsten.

Based on the approach on value chains above, it can be easily deduced that Europe is vulnerable to the supply of raw materials for all investigated technologies.

A similar analysis was carried out for the strategic sectors of the EU industry that rely on renewable energy, electric mobility, defense and aerospace industries, which potentially face supply risks for a number of different raw materials [51] (Figure 1).

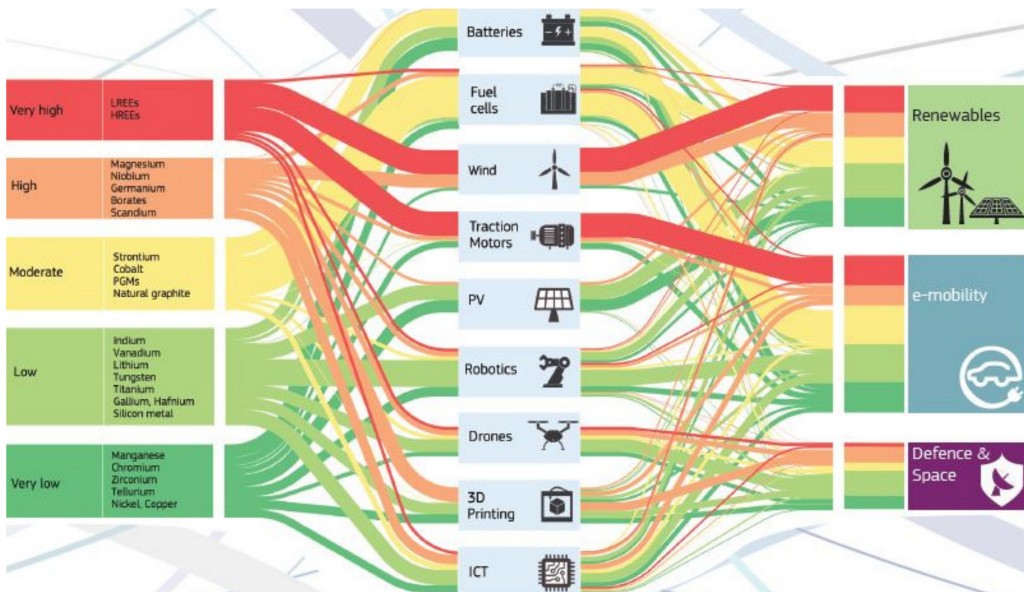

**Figure 1.** The supply risk of CRMs (according to 2020 criticality assessment [44]) and other raw materials in strategic values chains and strategic industrial sectors in the EU [51].

Technologies used in the renewable energy sector involve not only wind turbines and photovoltaics (for energy generation), but also rechargeable batteries and fuel cells (for energy storage), as well as robotics, 3D printing, and digital technologies (in the manufacturing and conversion, as well as transmission of electricity via smart grids). As a result, the critical raw materials indispensable to the value chain of renewable energy include boron and borates, cobalt, dysprosium (HREE), gallium, germanium, graphite, indium, magnesium, niobium, neodymium and praseodymium (LREE), platinum group metals, and silicon metal [51].

Additionally, the development of e-mobility in the EU will require the deployment of multiple new technologies, including batteries, fuel cells, traction motors, and digital technologies. This will, consequently, influence the demand for critical raw materials such as REEs (Nd, Dy, Pr) and boron for motors in EVs [49], as well as lithium, cobalt, natural graphite for energy storage in LiBs [68], and platinum in FCs. Lightweight parts of vehicles will require CRMs, such as magnesium, niobium, silicon metal and titanium, while electronic components need gallium, germanium, and indium [51].

The EU is largely dependent on imports of 13 of the 39 raw materials utilized in the defense sector, including CRMs such as boron (as borates), dysprosium (HREE), magnesium, neodymium, praseodymium, samarium and yttrium (LREE), niobium, tantalum, and titanium [2]. Overall, for more than two thirds of those raw materials, the share of imports exceeds 50% [51].

The analysis of the mineral CRMs utilized in key technologies showed that the majority of them (16 of 29) are indispensable for at least two the EU's strategic industrial sectors (Table 2).

### 3.3. Possible Sources of Critical Raw Materials in the EU

The majority of primary raw materials essential for the EU's economic growth are produced and supplied by non-European countries. This is the case for antimony, beryllium, bismuth, borates, molybdenum, niobium, PGMs, phosphorus, REEs, tantalum, titanium, vanadium, and zirconium [46]. This is either due to the absence of mineral deposits in the EU, or—even more often—to economic, environmental, and/or societal constraints in the exploration and extraction activities (closure of existing mines, resistance to launching new mines, limited access to the land due to existing infrastructure, etc.). Furthermore, the

largest and more profitably exploited deposits are found overseas, while deposits existing in the EU are usually small and hard-to-exploit.

Many national and EU-funded mineral-related projects [53,59–64] have highlighted Europe's mineral resources of some CRMs, including those with a current 100% import dependency, as antimony (with resources estimated, e.g., in Austria, Bulgaria, Finland, France, Italy, Slovakia, Sweden), bauxite (e.g., Greece, France, Hungary, and Romania), lithium (e.g., Sweden, Finland, Spain, France, the Czech Republic), magnesium (e.g., Austria, Bulgaria, Greece, Ireland, Italy, Poland, Slovakia, Spain), and REEs (e.g., Finland, France, Greenland, Portugal, and Sweden). Major European mineral belts, such as the Fennoscandian, Iberian, and Carpathian-Balkan, have been found as likely exploration targets for new resources that might be susceptible to being mined [53,54]. However, the secure and sustainable supply of some raw materials from the EU own primary sources is not only a matter of geological constraints but also of land accessibility, national legislative constraints, or social rejection [70]. As a result, despite significant potential for developing projects to source critical raw materials, the EU has been focused on processing imported raw materials rather than using domestic primary sources.

Data on the mineral resources and reserves at a global scale, including those in some European countries, are reported on an annual basis by the US Geological Survey; however, they are often incomplete and not updated [54,55,60]. Data on the European mineral deposits and their resources are available in various databases and publications; they have been also collected during past EU research projects [53,59,60]. Some data concerning resources and reserves of raw materials have been presented in the European Minerals Yearbook [71]. However, these data were reported based on various national/regional codes not always consistent with the United Nations Framework Classification (UNFC) system or the CRIRSCO Template [67]. Therefore, it is not possible to determine the total volume of resources and reserves in individual countries and the entire EU. Moreover, the evaluation of these resources is often based on historic estimates and some of these occurrences were of little economic interest.

Nevertheless, despite the data inconsistency, some mineral raw materials for which resources are reported in the EU can be indicated. Of the 16 mineral raw materials required in at least two industrial sectors which are strategic for the EU (see Section 3.2), the most promising in view of current or future mining production, with proven or estimated mineral resources, are 11, i.e., cobalt, graphite (natural), lithium, magnesium, niobium, PGMs, REEs (HREE and LREE), silicon metal, titanium, and tungsten (Table 3). For the remaining critical mineral raw materials, the EU's resources/reserves are not known, or relevant data are not available. However, except for borates, they occur merely in trace amounts in deposits whose exploitation targeted other metals. Therefore, gallium, germanium, indium, and vanadium can be recovered mainly as by-products [65,66]. The potential sources for these metals are indicated in Table 3.

Among these 11 mineral raw materials, there are two that could be mined in the Union in quantities large enough to meet its consumption needs for the next few decades. These are minerals suitable for the production of silicon metal (high purity quartz and quartz sand) and magnesium (high purity magnesite or dolomite, magnesia brines). Their European resources are abundant, but the cost of energy makes their production very expensive and limited to Norway and Iceland, where electricity generation is based on hydropower or geothermal energy, although they do not belong to the EU.

## 4. Discussion

Among 30 critical raw materials for the EU, identified in the list updated in 2020 [44], there are only two for which the EU countries are the global leading suppliers: hafnium (France) and strontium (Spain) [45,51]. Additionally, many construction and industrial minerals, such as aggregates, feldspar, gypsum, kaolin clay, limestone (also high purity), magnesite, perlite, silica sand, and sulfur, are exploited in sufficient quantities in the EU in order to avoid significant foreign imports. However, these are nothing but exceptional

cases [71]. The EU is largely dependent on imports of almost all CRMs utilized in strategic technologies, especially in renewable energy systems, and, therefore, is exposed to a high risk of supply disruption [51]. Currently, major world players in CRMs production are: China (REEs—86%, gallium, germanium and bismuth—80% each, antimony—74% of global supplies), Brazil (niobium—92%), the United States (beryllium—88%), South Africa (platinum group metals—71–93%), Congo (cobalt—60%) [47,65].

The EU is currently producing only 1% of all battery raw materials overall (with another 3% coming from rest of Europe) [3,51]. It should be mentioned, however, that the Union is significantly investing along the entire battery value chain that resulted, e.g., in ongoing construction of two huge fully European factories for battery cell production, namely Northvolt Ett in Sweden and MES HE3DA in the Czech Republic. In addition, leading Asian and US companies (Tesla, LG Chem, and CATL) are investing in the production capacity in Europe. If all the announced (25) projects in Li-ion factories are realized, this will add a total of approximately 500 GWh of production capacity for Europe in 2030. It is foreseen that Europe will have a 16% share of the 2550 GWh global battery market by 2029 compared to just less than 6% of today's 450 GWh [72]. The most likely result will be skyrocketing demand for mineral raw materials required in the battery value chain, especially cobalt, lithium, and graphite.

When it comes to renewable energy and e-mobility technologies, one of the most concerning features is the supply risk of the REEs in permanent magnets (PM) generators [49]. The European share in sourcing REEs for this technology is negligible (1%, while the EU's is 0%) [48]. The same feature refers to electric traction motors with permanent magnets—especially NdFeB—which are expected to dominate the market in the future. The EU produced 6% of the raw materials required in photovoltaics, 2% in robotics, and merely 1% in 3D printing technology [49–51].

Numerous CRMs are subject to trade restrictions such as export tariffs and export quotas, licensing requirements, and other measures, especially for trade in high-grade crude ores and concentrates, and other non-processed materials [73,74]. In the past, there were also reinforced concerns related to banning the exportation of some CRMs, e.g., REEs from China to Japan (in 2010) [47] and cobalt from the DRC. CRMs have also been an issue in the recent trade dispute between the USA and China. From 2000 onwards, China became the world's leading producer and exporter, and quickly achieved near monopoly control over the REE supply chain. The dominant market position of China leads to recurring supply risks. Highly concentrated production of some raw materials contributes to the fact that restrictions on trade in such materials are particularly disruptive to the supply chains [47,48,75,76]. They result in fluctuations of raw material prices but also in uncertainty of supplies of raw materials for producers of semi-processed and final products [73]. As a consequence, this generates increased costs of manufacturing at all stages of the value chain, arising from multiple trading of some raw material before it is finally sold to the end-user [74]. Therefore, to secure supplies of raw materials for the various industry sectors of the EU, it is important to be able to assess the domestic potential for mineral resources and the feasibility of the exploitation of known resources, especially in the case of those of high economic significance.

The multi-stage analysis carried out in this paper made it possible to distinguish nine critical raw materials, which:

- Firstly, are essential for the EU economy and the risk of their supply disruption is high;
- Secondly, are recognized as important for at least two of the EU's strategic industrial sectors;
- Thirdly, there are some recognized, though often limited, resources in the EU, which are, or can be, utilized for the production of these raw materials. Launching and/or developing their production from primary sources may contribute to mitigating the EU's import dependency and risk of supply disruption.

The list of these identified raw materials includes: cobalt, graphite (natural), lithium, niobium, PGMs, HREEs and LREEs, titanium, and tungsten (Table 4). Among them,

some are scarce in the EU (at least in terms of supply from primary sources of favorable mineral deposits), whereas the internal supply meets a part of the EU's demand in other materials. The first group includes lithium (only ceramic grades produced in Portugal), niobium, HREEs and LREEs, and titanium. The EU is largely deficient in PGMs, with marginal supplies coming from Finnish and Polish mining operations. However, according to [53,77–79], a promising potential source for the recovery of PGMs (Pd ± Pt) could be porphyry Cu–Au deposits in Bulgaria and Greece. Nevertheless, in these cases it is very characteristic that one or two countries play a dominant role in meeting EU needs, e.g., Chile and Australia in deliveries of lithium, Brazil of niobium, Russia and South Africa of PGMs, China of HREEs and LREEs, while Norway, South Africa, and Canada are the main titanium providers [46,51,67].

Only in the cases of cobalt, graphite, and tungsten is part of the EU's demand (commonly up to 15–25%) supplied by its own deposits, i.e., cobalt from Finland, natural graphite from Austria, and tungsten from Portugal, Austria, and Spain. The remaining deliveries of cobalt come mainly from Congo, graphite from China, and tungsten from the USA, China, and Canada [51,67].

In evaluating the possibilities for launching or developing the selected CRMs production from EU primary sources, we considered their criticality (measured by EI, SR, and IR indexes), coupled with the availability of potentially economic indigenous resources. With regard to the degree of market monopolization among these nine mineral raw materials, the possibility of developing the EU's own deposits is focused on HREEs and LREEs, cobalt, niobium, lithium, and natural graphite. Their resource base (and sometimes also reserves) and the possibilities of its development within the EU are diversified in terms of both volume and quality [53–61,80,81].

According to [82], Europe has a wide range of REEs deposits and occurrences, which are assumed to constitute some potential in the discovery of further resources. The largest recognized reserves of REEs in the EU, exhibiting the highest potential for developing production, are known in alkaline igneous rock deposits in Sweden, with smaller resources in skarn deposits and iron oxide-apatite deposits, also in Sweden. Small carbonatite deposits are known in Finland and Germany, while small placer deposits of REEs are located in Spain [82,83]. However, there is a need to develop beneficiation and processing methods at the highest environmental and social standards in order to ensure their sustainable production.

Cobalt has been identified in 104 deposits, being recently explored in Europe, of which 79 are situated in Finland (with the only active mines producing cobalt in the EU, i.e., Kevitsa, Kylylahti, Terrafame) and Sweden. These are mainly polymetallic Ni-Cu-Zn-Co sulfide, Ni-Cu-PGE, and Au-Co deposits [53,84,85]. Other cobalt-bearing formations are located in the Kupferschiefer belt in Poland and Germany. However, the cobalt grade in these deposits is relatively low (ave. 0.005–0.008% Co) and not currently economically mineable without significant improvements in extraction technology. Some deposits in the Balkans and other countries could be also viable future sources of cobalt, as long as more efficient and cleaner processing technologies (e.g., hydrometallurgical) are implemented [53]. These include almost thirty cobalt-bearing lateritic nickel ore deposits and mineral occurrences in Greece (some are currently exploited by Larco) [46,86].

The distribution of lithium in Europe shows the promising Li potential of the Variscan belt of south and central Europe in deposits of various types, i.e., pegmatites (including the largest pegmatite-hosted lithium resource in Europe, i.e., Sepeda in Portugal), rare-metal granites (Beauvoir in France), greisens and quartz veins (Cinovec in the Czech Republic), and Li-rich geothermal brines (Insheim in Germany) [67,87,88]. There have been six significant lithium projects in Finland, Austria, Spain, and Portugal (with the most promising project being Mina de Barrosa), five of which planned to have integrated mining and refining operations. Portugal's high level of renewable energy supplies (60%), combined with relatively low costs of labor and energy, should help the deposit development [89].

Graphite deposits of economic importance are rare within the EU. The bulk of graphite concentrations have been found in Fennoscandia and Austria, with active mines situated in this country (Kaisersberg amorphous graphite mine) [87,90]. Additional graphite resources have occurred in Sweden (Vittangi mine starting its operation in 2021, possible reopening the Woxna mine) and Germany [53,91].

The least likely material to be supplied from domestic sources of the EU is niobium, as the only relevant sources for it are P-Fe-Nb deposits in Finland, and some small individual deposits in Portugal and Spain [46,53].

The potential production of Ga, Ge, In, and V as by-products will heavily depend on appropriate technology for ore processing and on the demand for the main commodities with which these are associated (W, Pb, Zn, Ti, bauxite). Further, the production of by-product primary gallium was stopped in the last two operating installations in Hungary (2013) and Germany (2016) due to the cheaper recovery of this element from industrial waste. Germanium had been extracted from leaching residues of a zinc refinery in Finland that processed imported ore up to 2015. Since 2016, Ge production in Finland has been insignificant (probably due to economic reasons). Only indium is recovered in Belgium and France from zinc ore concentrates imported from third countries.

Although Europe has abundant vanadium resources in Fe-Ti ore deposits in Scandinavia and Poland, their mining remains not economically viable [45]. However, the EU dependency on vanadium imports can be mitigated by the Skåne project located in southern Sweden (shale-hosted Hörby deposit [92]), and by a possible reopening of the abandoned mines in Mustavaara (Finland) [53].

It should be underlined that the information on mineral resources and occurrences in the EU, collected from different sources, are sometimes ambiguous, incomplete, or misleading. For example, no significant sources of gallium are identified in the EU; although, according to the updated map of the CRMs deposits [55], there are some deposits in Poland in which the occurrence of this element is reported. Gallium and germanium in Poland are found in an undeveloped Zn-Pb deposit as trace elements (resources 0.13 kt Ga and 0.03 kt Ge [93]). In fact, gallium and germanium have never been recovered, despite the extraction and processing of Zn-Pb ores containing these metals.

It should also be mentioned that the extraction (or recovery as by-product) of some CRMs in the EU (e.g., graphite, gallium, germanium) has declined or ceased due to the intense competition of cheaper supplies from third countries, especially China. Another important impediment has been the EU's Regulation on the Registration, Evaluation, Authorization and Restriction of Chemicals (REACH), which came into force in 2007 [94]. This regulation imposed limitations on the production and trade of numerous mineral raw materials and mineral-based compounds, negatively affecting their consumption within the EU.

## 5. Conclusions

As the demand for raw materials in the EU is projected to double by 2050, their diversified sourcing is an essential objective for the EU [95]. Therefore, the development of the raw materials' production from local sources seems to be crucial in view of the future needs of industry in the EU. The evaluation of the EU's mineral resources and the feasibility of their exploitation should be a key issue in the currently ongoing debate on the security of supply of raw materials that are important for the development of the EU's economy and related risk mitigations [95–98]. This paper is a contribution to the assessment of the EU's mineral resources' potential for launching (developing) the production of critical raw materials. The CRMs indicated in this study are cobalt, graphite (natural), lithium, niobium, PGMs, HREEs and LREEs, titanium, and tungsten. They are required in at least two strategic industry sectors, including renewable energy (niobium, REEs) and e-mobility (cobalt, natural graphite, lithium, niobium, PGMs, REEs, titanium, and tungsten), as well as defense and aerospace (cobalt, natural graphite, lithium, niobium, PGMs, REEs, titanium, and tungsten).

However, it should be acknowledged that increased domestic raw materials production will not be sufficient to meet the demand of high technologies. The EU will remain dependent on third countries to obtain some CRMs in sufficient quantities, i.e., borates, germanium, niobium, PGMs, and REEs, whose external supplies can be constrained and disrupted as long as their production is ecologically and socially problematic. In our opinion, promising approaches to mitigate EU dependency and to reduce supply risk disruptions may include:

- Promoting the use of secondary raw materials and improving recycling rates of electronic waste (e.g., to obtain REEs and some other CRMs), coupled with restrictions on exports of electronic scrap to Asia (especially China) or Africa;
- Developing the recovery of accompanying elements contained in the raw materials imported to the EU, e.g., REEs from imported phosphate rock, gallium, germanium, or indium from imported concentrates of polymetallic ores;
- The assessment of waste sources (old slags and dams) for the recovery of CRMs (e.g., REEs from phosphogypsum);
- Considering re-starting some domestic CRMs operations;
- A geological (re)assessment of known deposits and metallogenic provinces within the EU, with feedback from academia, industries, and governmental agencies;
- Securing CRMs supply from non-EU European countries, where large reserves of some CRMs are known, e.g., graphite, cobalt, REEs in Norway [54,95,98], REEs in Greenland [82], and graphite and titanium in Ukraine [98], etc.;
- Promoting exploration surveys in foreign countries with which the EU may establish raw materials partnerships to diversify the sources of its supply;
- Prioritizing the rational and effective use of raw materials (in line with circular economy approach).

Finally, it should be emphasized that secure supplies of raw materials from domestic sources are considered to be a starting point for the EU value chains from upstream to downstream. Therefore, it is also important to develop in-house processing (refining, etc.) of the raw materials that will possibly be mined in the EU, instead of exporting them. Otherwise, the EU will become increasingly dependent on imports of value-added mineral products.

**Author Contributions:** Conceptualization, E.L., K.G. (Katarzyna Guzik) and K.G. (Krzysztof Galos); methodology, E.L., K.G. (Katarzyna Guzik) and K.G. (Krzysztof Galos); investigation, E.L., K.G. (Katarzyna Guzik) and K.G. (Krzysztof Galos); data curation, E.L., K.G. (Katarzyna Guzik) and K.G. (Krzysztof Galos); writing—original draft preparation, E.L., K.G. (Katarzyna Guzik) and K.G. (Krzysztof Galos); writing—review and editing, E.L. All authors have read and agreed to the published version of the manuscript.

**Funding:** This research was supported by the European Union's Horizon 2020 project MINLAND—Mineral Resources in Sustainable Land Use Planning (grant agreement number 776679). The preparation and publication of this paper has been supported by the Polish National Agency for Academic Exchange under Grant No. PPI/APM/2019/1/00079/U/001.

**Institutional Review Board Statement:** Not applicable.

**Informed Consent Statement:** Not applicable.

**Data Availability Statement:** Not applicable.

**Acknowledgments:** Authors are very grateful to the four anonymous reviewers for their constructive comments on the first submission of this paper.

**Conflicts of Interest:** The authors declare no conflict of interest.

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
