# Peer review of "On the Possibilities of Critical Raw Materials Production from the EU’s Primary Sources"

_resources, doi:10.3390/resources10050050_

Round 1

Reviewer 1 Report

Dear authors,

Here is my review on the Manuscript ID: resources-1185975, “ On the Possibilities of the Critical Raw Materials Production from the EU’s Primary Sources, by Ewa Lewicka et al.

This contribution has evaluated mineral raw materials characterized as critical for the EU and the possibilities of their supply from the EUs primary sources, including an assessment of for the EU economic growth, the significance for strategic industrial sectors and the potential availability from the EU’s mineral deposits. This manuscript is well organized and written. In my opinion this article is suitable for publication to the Resources journal after minor revision.

Specific comments

I think that the Introduction could be shorted, in order to avoid repetition with the section 2. Materials and Methods.

Lines 606-607: …. Some deposits in Balkans and other countries could be also viable future sources of cobalt provided that new processing technologies  (e.g., hydrometallurgical) are implemented [54]

Please, check again the cited ref [54], you could re-write the provided information concerning Greece:  “…The lateritic nickel ores of exploited by Larco (Apostolikas & Kountourellis 2014) contain cobalt, which is extracted from the nickel ore and used in the ferronickel production.”

Also, “…Almost thirty cobalt-bearing ore deposits and mineral occurrences are known in Greece. Cobalt reserves reported for the deposits in Greece comprise almost 50,000 t of Co from, mostly, lateritic Ni deposits. Mineral resources include additional 79,000 t of Co.  In addition, the level of the Co content in Fe-Ni laterites of the Balkan Peninsula is available in relative publications.

Also, I think that the provided information concerning the platinum-group elements (PGE) in the Balkan Peninsula is missing. Again, in the cited ref [54] it is written: “The Skouries porphyry Cu deposit is reported to be enriched in PGM (John & Taylor 2016). The Skouries 2 ore body is reported to have 23.1 t Pt+Pd in 568 Mt of ore (average grade 0.047 g/t Pt+Pd; John & Taylor 2016).  In addition, I suggest the following publications (and references therein), emphasizing that porphyry-Cu-Au systems have the potential for (Pd±Pt) production, as a by-product, the main products being Cu and Au:

Augé, T.; Petrunov, R.; Bailly, L. On the mineralization of the PGE mineralization in the Elastite porphyry Cu–Au deposit, Bulgaria: Comparison with the Baula-Nuasahi Complex, India, and other alkaline PGE-rich porphyries. Can. Mineral. 2005, 43, 1355–1372.

Economou-Eliopoulos, M., 2005. Platinum-group element potential of porphyry deposits. In: Mungall, J.E. (Ed.), Exploration for Platinum-group Element Deposits. Mineral. Association of Canada, Short Course 35, pp. 203–246. Wang, M. et al. 2020.

Factors controlling Pt–Pd enrichments in intracontinental extensional environment: Implications from Tongshankou deposit in the Middle–Lower Yangtze River Metallogenic Belt, Eastern China. Ore Geology Reviews 124 (2020) 103621.

Best wishes

Reviewer 2 Report

The introduction is too long and presents information that is found in other sections of the paper as well. All repetitive passages can be omitted or drastically shortened so that the authors may avoid repetition. Repetition in this paper can be even nightmarish at times, as the authors resort abundantly to it —that must not be!

It seems as though the authors rushed to present their analysis in the introduction, which is not the place to do it. They may well include their general ideas or parameters succintly there and fully develop them later on.

In general, this manuscript is too wordy: the same can be achieved by "shaving it up" at about a 15-20%. Sometimes, its wordiness is a real issue because it makes reading it boring or misleading and it makes easier to lose the stream of thought. The sentences must be shorter and more to the point, ever choosing the simplest wording possible. All in all, a significant shortening is a must. Most long and useless phrasing must be cut down, and the authors must also avoid using the Saxon genitive " 's" in a formal text.

Besides wordiness, this paper is somewhat messy: the authors should clearly separate the results obtained in their analysis, the subsequent discussion, and some concluding remarks. The conclusions must be short and clear, and not bound to the discussion. I have made many annotations on the manuscript that should help the authors to get through rewriting.

Another problem is that I do not see conclusions but the few remarks that can be made intrinsically to this analysis. In other words, the conclusions should go farther than the immediate results in the analysis and should push through the issue by trying to answer key questions. It is clear that the EU is not self-sufficient in any critical raw material, but in which its production is relevant? That is already stated in the paper, but the authors should make clear and short statements that go to the point. Then, in which materials is the EU deemed to remain dependent on third countries? This is more or less addressed, but there is no diagnostic beyond it, in the sense that the authors should have an opinion by now of how to mitigate the effects of such dependence. After all, they work at the Mineral and Energy Economy Research Institute of the PAS. I need to emphasize that this paper only provides a diagnosis, but it prescribes absolutely nothing. We should expect some proactivity on the matter. Which can be possible solutions for the issue? To lower the environmental restrictions in the EU? To promote aggressively exploration surveys or the enhancement of metallurgical processes? To promote exploration surveys on countries with which the EU may establish better deals than with the current monopolizers of some raw material stock? Or is there no solution? Unless the authors do not address properly such issues, their paper will be rendered useless because it ends as it started, with the idea that the EU is at the mercy. Let me insist: no diagnosis is worthwhile unless it is followed by treatment. The process that leads the authors to decide which would be, in their view, the best possible treatment is what we would call here "discussion". Real discussion elements are tattered here and there, and they could make a good start once the authors succeed to separate the presentation of data (the results of their analysis), the discussion, and the conclusions.

There are some geopolitic problems as well: this paper includes in the analyses countries or territories that are not EU members, as Norway or Greenland. I think that these should be left off the analysis out of respect to their citizens, which have democratically chosen to stay away of the club. If this analysis relies on their resources, however partly, it is a faulty analysis. If these countries were to be analyzed along with all EU members just because they are conterminous countries to the EU, then why not including Russia, Ukraine, Serbia, the UK, etc.? Would it not be nice to include the Trepça mines from Kosovo as well? Well, of course Greenland is under Danish rule for the time being and contains virtually 100% of the mineral resources under such rule, but that should not be the point. The new EU haughtiness does not stray far from the old colonialism. So please leave Norway and Greenland alone unless you are able to exhibit some agreement between such parties and the EU that allows them to be considered as "second level" near-domestic purveyors, but specify very clearly that they must not be considered as domestic at any rate.

Author Response

Response to Reviewer 2 Comments

Question 1: The introduction is too long and presents information that is found in other sections of the paper as well…, a significant shortening is a must. Most long and useless phrasing must be cut down, and the authors must also avoid using the Saxon genitive " 's" in a formal text.

Response 1: The introduction has been shortened. We did our best also to shorten the article and to reduce wordiness. However, because other Reviewers asked for some additional information in the introduction (referring to methodologies of criticality assessment in the USA, Japan and the UK, as well as to calculation of EI and SR indexes in the EU), the shortening is probably not as spectacular as expected. The Saxon genitive ‘s’ has been removed everywhere. We appreciate very much the Reviewer annotations on the manuscript that help us to rewrite and edit the text in English. We have accepted all suggested corrections. The comments on the pdf version of the manuscript also have been responded.

Question 2: It seems as though the authors rushed to present their analysis in the introduction, which is not the place to do it. They may well include their general ideas or parameters succintly there and fully develop them later on.

Response 2: The paragraphs referring to the analysis were removed to avoid repetition.

Question 3: …the authors should clearly separate the results obtained in their analysis, the subsequent discussion, and some concluding remarks.

Response 3: The results have been separated from the discussion, while the conclusions have been presented in the separate section.

Question 4: Another problem is that I do not see conclusions but the few remarks that can be made intrinsically to this analysis. In other words, the conclusions should go farther than the immediate results in the analysis and should push through the issue by trying to answer key questions. It is clear that the EU is not self-sufficient in any critical raw material, but in which its production is relevant? That is already stated in the paper, but the authors should make clear and short statements that go to the point. Then, in which materials is the EU deemed to remain dependent on third countries? This is more or less addressed, but there is no diagnostic beyond it, in the sense that the authors should have an opinion by now of how to mitigate the effects of such dependence.

Response 4: Our opinion on above questions is expressed in the conclusions:

‘As the demand for raw materials in the EU is projected to double by 2050, their diversified sourcing is an essential objective for the EU [79]. Therefore, the development of the raw materials production from local sources seems to be crucial in view of the future needs of industry in the EU [80]. The evaluation of the EU mineral resources and feasibility of their exploitation should be a key issue in the currently ongoing debate on the security of supplies of raw materials that are important for the development of the EU economy and the related risks mitigation [69,70,79,81]. This paper is a contribution to the assessment of the EU mineral resources potential for launching (developing) production of critical raw materials. The CRMs indicated in this study are cobalt, graphite (natural), lithium, niobium, PGMs, HREEs and LREEs, titanium and tungsten. They are required in at least two strategic industry sectors, including renewable energy (niobium, REEs), e-mobility (cobalt, natural graphite, lithium, niobium, PGMs, REEs, titanium and tungsten) as well as defense & aerospace (cobalt, natural graphite, lithium, niobium, PGMs, REEs, titanium and tungsten).

However, it should be acknowledged that increased domestic raw materials pro-duction will not be sufficient to meet the demand of high technologies. The EU will re-main dependent on third countries to get some CRMs in sufficient quantities, i.e., borates, germanium, niobium, PGMs, and REEs, which external supplies can be con-strained and disrupted while their production is ecologically and socially problematic. In our opinion promising approaches to mitigate the EU dependency and to reduce supply risk disruptions may include:

  • Promoting the use of secondary raw materials and improving recycling rates of electronic waste (e.g. to get REEs and some other CRMs), coupled with restrictions on exports of electronic scrap to Asia (especially China) or Africa,
  • Developing recovery of accompanying elements contained in the raw materials imported to the EU, e.g., REEs from imported phosphate rock, gallium, germanium, indium from imported concentrates of polymetallic ores,
  • The assessment of waste sources (old slags and dams) for the recovery of CRMs (e.g., REEs from phosphogypsum),
  • Considering re-starting of some domestic CRMs operations,
  • A geological (re)assessment of known deposits and metallogenic provinces within the EU with feedback from Academia, industry and governmental agencies,
  • Securing CRMs supply from non-EU European countries, where large reserves of some CRMs are known, e.g. graphite, cobalt, REEs in Norway [54,95,98,99], REEs in Greenland [91]), graphite and titanium in Ukraine [81], etc.,
  • Promoting exploration surveys in foreign countries with which the EU may establish raw materials partnerships to diversify sources of supply,
  • Prioritizing the rational and effective use of raw materials (in line with circular economy approach).

Finally, it should be emphasized that secure supplies of raw materials from domestic sources are considered to be a starting point for the EU value chains from upstream to downstream. Therefore, it is also important to develop in-house processing (refining, etc.) of the raw materials that will be possibly mined in the EU instead of exporting them. Otherwise, the EU will become increasingly dependent on imports of value added mineral products.’

Question 5: … this paper includes in the analyses countries or territories that are not EU members, as Norway or Greenland. I think that these should be left off the analysis out …

Response 5: Norway and Greenland, as non-members of the EU, have been deleted from the analysis. The resource potential of Norway have been considered in the paper because Norway is a part of EEA – European Economic Area, while Greenland is under Danish rule ‘for the time being’. We agree with the Reviewer that these countries shouldn’t be analyzed as domestic suppliers.

We are grateful for the Reviewer feedback and constructive comments that enable to improve the paper on the first submission.

Reviewer 3 Report

Thank you for submitting this work to Resources - I think your study is of great importance and I hope to see this being published shortly. I have some feedback that I like you to consider in order to make this a really impactful paper:

Abstract:

-not just green technologies also digitalization will required huge amounts of CRMs particularly REEs

-change to: "the critical raw materials (CRMs)"...

-what list of CRMs was used in the study, I think this is important to already mention here since the EU is constantly updating it

-present the results more clearly, this can be done better

Introduction:

-Line 27-36 maybe just provide the list of EU CRMs you considered here since it is such an important part of your study?

-you talk about import dependencies, this is important. You can make your arguments much stronger by quantifying this dependency. Maybe use the HHI as it was done previously for uranium (https://doi.org/10.1016/j.rser.2021.110740)

-could you quantify how much CRMs are needed for the EU green deal? I think this would again make your argument much much stronger

-what interesting strategies are used in the UK and Japan and the US? If you mention it here (which I think is great) you need to explain it. This could be a great chance to compare the different methodologies. Maybe in a table?

Materials & Methods:

-did you use PRISMA or some other reviewing methodology? I think this could be helpful and you could provide a convincing statement how your review is comprehensive. Otherwise this could come across as biased. You do not need to reference every paper you review. Reference only papers that you actually take data from.

Results:

line 191-192 is a direct repetition that was already mentioned in the intro

line 192-221 is material that should be in the intro and methods, and partly is. Make sure you really only present results here

how did you calculate the SR? that definition/equation should be provided in the methodology

Table 1:

explain: EI index, SR index

directly put yes/no in the table - that makes it easier to read and saves space

line 332-333 so now you use elemental symbols, that is great and you could consider doing that throughout the manuscript - most importantly it should be coherent what you do

Fig. 1

How was the supply risk calculated, how is this calculation different from your methodology? I would be very careful introducing other peoples data here since their analysis is probably very different from your own

Discussion:

-how realistic is it that we will see large mining operations in Europe again - I think this needs to be addressed here. Europe was a mining continent but companies go away since it is really hard to obtain a SLO. So I think your analysis that more mining is essential is good, but needs more discussion here how that can actually be realized

-I think mining from secondary resources should be discussed here. The EU does for instance have considerable amounts of Phosphogypsum (for instance in Poland) that could be mined for natural gypsum and REEs.

-CRMs could further be recovered from other waste streams such as REEs from electronic waste - we currently ship most of that to CHina where the CHinese do this. Another interesting topic you could discuss here

-The EU could further recover byproducts from primary imports such as REE from phosphate rock imports. This is already in consideration in the SeCREETs project in Norway. There is further a really good analysis on phosphate rock import flows into the EU that discusses the chances of recovering U from EU phosphate rock imports (https://doi.org/10.1016/j.resourpol.2019.02.012). So this could also contribute a good share I guess

-Lastly I would expect more analysis why import dependence is such an important factor - it intuitively makes sense, but some analysis would help to strengthen this aspect. Maybe a quick look into how china increase prices for REE some years ago would help.

Generally, I think this manuscript can profit from more quantitative data. The topic is really relevant and of high interest and I am sure this will be an impactful paper.

Reviewer 4 Report

The subject of the article is very interesting and the list of references is impressive. Although the aim of the manuscript is specified, it is not clear what is new in this paper. It is not sufficiently explained where is the progress beyond the state of the art and how the methodology applied in the current study of the manuscript differs from the methodology of the fourth EC communication on critical raw materials or EC Study on the EU’s list of Critical Raw Materials in obtaining the results in chapter 3.1.

The tables as well as figures given in the results are based on existing studies therefore it is not clear enough  what the authors' original work is. There is not much difference between the introduction and the results in the manuscript.

Moreover, the manuscript does not explain satisfactorily what are the EI index  and  SR index   and how are they calculated

Round 2

Reviewer 2 Report

I appreciate the effort of the authors in enhacing the manuscript and making it clearer and easier to read. Is is not the way in which I would have presented the case, but the authors must not be forced to adapt to the expectations or demeanour of the reviewers in matter of style and scope as long as their own actually meet the requirements of the journal. Let me take the liberty to state that, as a reviewer, my aim is to ensure the readability of some paper as well as its scientific soundness, from the representativeness of sampling to the conclusions. In other words, I am here to help if help is wanted, although help sometimes comes as a rough blow. My compliments the authors for their equanimity and thoroughness. It can be difficult to balance what ever all the reviewers have to say about some matter, and I think that the authors have done this quite beautifully.

I keep only one regret, though it is a big one. Being a geologist myself, I would have really liked to see more about geology and ore deposits in this paper. I know that that may be far from the area of expertise of the authors but, still, they could have allied themselves with an ore geologist. However, I reckon that, in such a case, they would have been told that they were merely digging an additional grave to a long buried corpse. Even so, such collaboration would have added the necessary geological context to this analysis and provided additional arguments to its purpose, no matter how grim the landscape would turn out to be. But, again, that was the choice the authors made —to me, a bad choice, but not as bad as to deserve a rejection.

I still think that there is room for some text shortening and "dismantling" wordiness for good. Therefore, I would encourage the authors to, yet again, go through the manuscript as if they were their own readers, not the authors, and get rid of any unnecessary wording. What ever one may shorten for the sake of clarity, must be shortened. That can only benefit the purpose of this paper.

From my side, I wish not to prolonge the case any farther. I think that, once the authors "shave the manuscript's whiskers" one last time, we should be ready to go.

Author Response

Response to Reviewer 2 Comments

Question 1: I appreciate the effort of the authors in enhancing the manuscript and making it clearer and easier to read. It is not the way in which I would have presented the case, but the authors must not be forced to adapt to the expectations or demeanour of the reviewers in matter of style and scope as long as their own actually meet the requirements of the journal. Let me take the liberty to state that, as a reviewer, my aim is to ensure the readability of some paper as well as its scientific soundness, from the representativeness of sampling to the conclusions. In other words, I am here to help if help is wanted, although help sometimes comes as a rough blow. My compliments the authors for their equanimity and thoroughness. It can be difficult to balance whatever all the reviewers have to say about some matter, and I think that the authors have done this quite beautifully.

Response 1: Thank you for your understanding. It has not been easy to meet the expectations of all the four Reviewers, but we did our best to face the challenge. We thank you again for your help (because help was wanted indeed) in enhancing the manuscript.

Question 2: I keep only one regret, though it is a big one. Being a geologist myself, I would have really liked to see more about geology and ore deposits in this paper. I know that that may be far from the area of expertise of the authors but, still, they could have allied themselves with an ore geologist. However, I reckon that, in such a case, they would have been told that they were merely digging an additional grave to a long buried corpse. Even so, such collaboration would have added the necessary geological context to this analysis and provided additional arguments to its purpose, no matter how grim the landscape would turn out to be. But, again, that was the choice the authors made —to me, a bad choice, but not as bad as to deserve a rejection.

Response 2: We have provided the geological information (as much as possible from various sources) in Table 3, hoping that this will be satisfactory background of the analysis. More discussion about geology and ore deposits would significantly increase the length of the article. We are aware that the revival of large mining operations in the EU is not realistic since it is hard to obtain a SLO, but we believe that improving CRMs supplies from domestic sources (primary and secondary) is possible, although probably in a scale far from the expectations.

  Question 3: I still think that there is room for some text shortening and "dismantling" wordiness for good. Therefore, I would encourage the authors to, yet again, go through the manuscript as if they were their own readers, not the authors, and get rid of any unnecessary wording. What ever one may shorten for the sake of clarity, must be shortened. That can only benefit the purpose of this paper.

Response 3: We get rid of or shortened/rephrased the following parts of the text:

Lines 93-95: The sentence (introduced to the original version of the manuscript) has been deleted: ‘Depending on the EU import reliance (IR), the two sets of the producing countries are taken into account — the global suppliers and the countries from which the EU is sourcing the raw materials.’

Lines 162-163: A part of the sentence (‘, i.e., in a rapidly developing new technologies and industry sectors’) has been deleted (repetition).

Lines 164-166: The sentence has been deleted. The following sentence begins with ‘A growing demand stems mainly from…’.

Lines 179-180: The sentence ’The dependency on imports of some CRMs may be reduced by developing their production from the EU resources.’ has been deleted.

Lines 201-206: A part of the paragraph has been deleted (‘The 29 mineral raw materials (excluding natural rubber) were thoroughly analyzed in respect to factors determining their criticality as well to possibilities of developing their production opportunities from primary sources in the EU member states. Simultaneously, the raw materials being crucial components in the EU’s strategic value chains - important for new technologies and industrial sectors development - have been identified.’).

Lines 224-228: A part of the paragraph has been deleted (‘On the other hand, CRMs for which there are no prospects for mining development within the EU due to a lack of or limited resources were distinguished. Complementary, the raw materials classified as critical in the EU but with a resource base large enough to meet even the world’s demand for the next decades, were also indicated.’).

Lines 363-364: The phrase ‘critical raw materials’ has been replaced by ‘CRMs’.

Line 376: The phrase ‘of the FCs’ has been deleted.

Line 397: A part of the sentence ‘Among the raw materials utilized’ has been deleted.

Lines 424-428: The sentence ‘The selection of the key technologies was based on anticipated growth rates leading to a notable increase in consumption of raw materials (e.g. wind and solar PV technologies), their relevance for strategic sectors such as defense or aerospace (e.g. 3D printing and drones) or importance across new emerging sectors (e.g. FC, robotics, digital technologies).’ has been deleted.

Lines 511-516: A part of the paragraph has been deleted (‘The ProMine project delivered a database of mineral occurrences and deposits divided into four groups depending on the resources volume of resources (very large, large, medium, small). The information on deposits of critical raw materials was compiled separately in order to encourage the mining industry to launch their extraction of new raw materials for the European manufacturing sector [55,56].’).

Lines 545-546: The sentence has been rephrased to: ‘Among above 11 mineral raw materials, there are two…’. A part of the sentence ‘that are important in at least two strategic EU industry sectors,’ has been deleted.

Lines 605-607: A part of the text has been rephrased ‘When it comes to the renewable energy and e-mobility technologies, one of the most concerning feature is the supply risk of the REEs in permanent magnets (PM) generators [49]. The European share in sourcing REEs for this technology is negligible…’.

Lines 610-612: The sentences ‘The share of the EU in the raw materials supplies for PMs accounts for 1%. In the EU-members produce 6%, contribute to the supply of the raw materials utilized.’ have been deleted.

Lines 612-613: The part of the text has been rephrased as follows ‘The EU produced 6% of the raw materials required in photovoltaics, 2% - in robotics, and merely 1% - in 3D printing technology [49,50,51].’

Lines 613-614: The sentences deleted ‘Nineteen of the 44 raw materials utilized in robotics were flagged as critical to the EU economy. The supply from European countries is negligible (1%).’ (repetition).

Line 748: To avoid repetition ‘Austria’ has been replaced by ‘this country’.

Line 793:  To avoid repetition ‘in Poland’ has been deleted.

Lines 795-799: The part of the text has been rephrased as follows ‘It should be also mentioned that extraction … of some CRMs in the EU (e.g., graphite, gallium, germanium) has declined or ceased due to intense competition of cheaper supplies from third countries, especially China.’

Question 4: From my side, I wish not to prolong the case any farther. I think that, once the authors "shave the manuscript's whiskers" one last time, we should be ready to go.

Response 4: This is what we hope.

We thank you again for your support in enhancing our paper clarity. We believe that now, after ‘shaving the manuscript’s whiskers’ it will deserve for publication.

Reviewer 3 Report

Well done - looking forward to seeing this published soon

Author Response

Dear Reviewer,

Thank you for your constructive comments and support.

Kind regards,

Ewa Lewicka

Reviewer 4 Report

The reviewer’s comments have been duly replied. The manuscript is of higher quality and more comprehensive after all the corrections and improvements have been made.

Author Response

Dear Reviewer,

Thank you for the acceptance of our paper publication.

Kind regards,

Ewa Lewicka